# Clinical Course and Severity of COVID-19 in 940 Infants with and without Comorbidities Hospitalized in 2020 and 2021: The Results of the National Multicenter Database SARSTer-PED

**DOI:** 10.3390/jcm12072479

**Published:** 2023-03-24

**Authors:** Małgorzata Pawłowska, Maria Pokorska-Śpiewak, Ewa Talarek, Anna Mania, Barbara Hasiec, Elżbieta Żwirek-Pytka, Magdalena Stankiewicz, Martyna Stani, Paulina Frańczak-Chmura, Leszek Szenborn, Izabela Zaleska, Joanna Chruszcz, Ewa Majda-Stanisławska, Urszula Dryja, Kamila Gąsiorowska, Magdalena Figlerowicz, Katarzyna Mazur-Melewska, Kamil Faltin, Przemysław Ciechanowski, Michał Peregrym, Joanna Łasecka-Zadrożna, Józef Rudnicki, Barbara Szczepańska, Ilona Pałyga-Bysiecka, Ewelina Rogowska, Dagmara Hudobska-Nawrot, Katarzyna Domańska-Granek, Adam Sybilski, Izabela Kucharek, Justyna Franczak, Małgorzata Sobolewska-Pilarczyk, Ernest Kuchar, Michał Wronowski, Maria Paryż, Bolesław Kalicki, Kacper Toczyłowski, Artur Sulik, Sławomira Niedźwiecka, Robert Flisiak, Magdalena Marczyńska

**Affiliations:** 1Department of Infectious Diseases and Hepatology, Faculty of Medicine, Collegium Medicum, Nicolaus Copernicus University, 85-067 Bydgoszcz, Poland; 2Department of Children’s Infectious Diseases, Medical University of Warsaw; Regional Hospital of Infectious Diseases in Warsaw, 02-091 Warsaw, Poland; 3Department of Infectious Diseases and Child Neurology, Poznan University of Medical Sciences, 61-701 Poznań, Poland; 4Department of Children’s Infectious Diseases, Provincial Jan Boży Hospital in Lublin, 20-400 Lublin, Poland; 5Department of Pediatrics and Infectious Diseases, Wroclaw Medical University, 50-367 Wrocław, Poland; 6Department of Pediatric Infectious Diseases, Medical University of Lodz, 90-419 Łódź, Poland; 7Department of Pediatrics and Infectious Diseases, Regional Hospital in Szczecin, 71-455 Szczecin, Poland; 8Collegium Medicum, Jan Kochanowski University, 25-369 Kielce, Poland; 92nd Department of Pediatrics, Centre of Postgraduate Medical Education, 01-813 Warsaw, Poland; 10Department of Pediatrics and Neonatology with Allergology Center, Central Clinical Hospital of the Ministry of the Interior, 02-507 Warsaw, Poland; 11Department of Pediatrics with Clinical Assessment Unit, Medical University of Warsaw, 02-091 Warsaw, Poland; 12Department of Pediatrics, Pediatric Nephrology and Allergology, Military Institute of Medicine, 01-755 Warsaw, Poland; 13Department of Pediatric Infectious Diseases, Medical University of Bialystok, 15-089 Białystok, Poland; 14Department of Pediatric Infectious Diseases, Pomeranian Center of Infectious Diseases and Tuberculosis in Gdańsk, 80-214 Gdańsk, Poland; 15Department of Infectious Diseases and Hepatology, Medical University of Bialystok, 15-089 Białystok, Poland

**Keywords:** coronavirus disease 2019 (COVID-19), infants, severe acute respiratory syndrome coronavirus 2 (SARS-CoV-2), variants of concern (VOCs)

## Abstract

This study aimed to analyze the differences in severity and clinical characteristics of COVID-19 in infants hospitalized in Poland in 2021, when the dominance of variants of concern (VOCs) alpha and delta was reported, compared to 2020, when original (wild) SARS-CoV-2 was dominant (III–IV vs. I–II waves of the pandemic, respectively). In addition, the influence of the presence of comorbidities on the clinical course of COVID-19 in infants was studied. This multicenter study, based on the pediatric part of the national SARSTer database (SARSTer-PED), included 940 infants with COVID-19 diagnosed between March 1, 2020, and December 31, 2021, from 13 Polish inpatient centers. An electronic questionnaire, which addressed epidemiological and clinical data, was used. The number of hospitalized infants was significantly higher in 2021 than in 2020 (651 vs. 289, respectively). The analysis showed similar lengths of infant hospitalization in 2020 and 2021, but significantly more children were hospitalized for more than 7 days in 2020 (*p* < 0.009). In both analyzed periods, the most common route of infection for infants was household contact. There was an increase in the percentage of comorbidities, especially prematurity, in children hospitalized in 2021 compared to 2020. Among the clinical manifestations, fever was predominant among children hospitalized in 2021 and 2020. Cough, runny nose, and loss of appetite were significantly more frequently observed in 2021 (*p* < 0.0001). Severe and critical conditions were significantly more common among children with comorbidities. More infants were hospitalized during the period of VOCs dominance, especially the delta variant, compared to the period of wild strain dominance, even though indications for hospitalization did not include asymptomatic patients during that period. The course of COVID-19 was mostly mild, characterized mainly by fever and respiratory symptoms. Comorbidities, particularly from the cardiovascular system and prematurity, were associated with a more severe course of the disease in infants.

## 1. Introduction

Severe acute respiratory syndrome coronavirus 2 (SARS-CoV-2), the etiologic agent of coronavirus disease 2019 (COVID-19), rapidly spread worldwide, causing a global pandemic. According to the data published by the National Institute of Public Health NIH—National Research Institute, there were 2,834,287 cases of COVID-19 reported in Poland in 2020 and 1,289,293 in 2021 [1]. The proportion of infected children remains unknown. Although the effects of COVID-19 have been more significant in adults, children are also infected with SARS-CoV-2, and COVID-19 can lead to severe outcomes in pediatric patients, such as severe multisystem inflammatory syndrome in children (MIS-C) [2]. According to the study by Nikolopoulou et al., serologic surveys indicate that half of the children who tested positive for SARS-CoV-2 reported no symptoms, and children with COVID-19 are at lower risk of hospitalization and life-threatening complications compared to adults [3].

Our previous analysis of the clinical course of COVID-19 in 300 infants, selected from 1283 children diagnosed with COVID-19 between March and December 2020, showed that COVID-19 in infants usually manifests as a mild gastrointestinal or respiratory infection, but pneumonia is also observed with falls in oxygen saturation, requiring oxygen therapy. Gastrointestinal symptoms are common in infants infected with SARS-CoV-2, and loss of appetite in infants may lead to hospitalization [4].

Symptoms associated with SARS-CoV-2 infection are generally milder in children than in adults. In addition, several risk factors for severity have been identified. Schober et al., who performed multivariable ordinal logistic regression analyses adjusted for age, chest imaging findings, laboratory-confirmed bacterial and/or viral coinfection, and MIS-C diagnosis in 403 hospitalized children in a median age of 3.78 years, revealed that the presence of comorbidities, obesity, and chromosomal disorders are independent risk factors for COVID-19 severity. Age was not an independent risk factor, but different age-specific comorbidities were associated with more severe disease in age-stratified adjusted analyses: cardiac and non-asthma pulmonary disorders in children <12 years old and obesity in adolescents ≥12 years old. Among infants < 1 year old, neurological and cardiac disorders were independent predictors of severe disease [5].

Since the start of the SARS-CoV-2 pandemic, children aged ≤12 years have always been defined as underrepresented in terms of SARS-CoV-2 infections’ frequency and severity [6,7]. Increased transmissibility across all age groups has been reported for SARS-CoV-2 variants of concern (VOCs), including the delta and omicron variants [8]. According to our recent observations, it seems that infants may play a particular role in the transmission of SARS-COV-2 infections in households, despite mild or asymptomatic courses [4].

Alteri et al., correlating SARS-CoV-2 transmission dynamics with clinical and virological features in 612 SARS-CoV-2 positive patients aged ≤12 years, demonstrated a sizeable circulation of different SARS-CoV-2 lineages over the four pandemic waves in the pediatric population, sustained by local transmission chains. They revealed that age <5 years, the highest viral load, and gamma and delta clades positively influence this local transmission. No correlations between COVID-19 manifestations and lineages or transmission chains are seen, except for a negative correlation between B.1.1.7 and hospitalization [9].

This study aimed to analyze the differences in severity and clinical characteristics of COVID-19 in infants hospitalized in Poland in 2021, when the dominance of VOCs alpha and delta was reported, compared to 2020, when original (wild) SARS-CoV-2 was dominant (III–IV vs. I–II waves of the pandemic, respectively). In addition, the influence of the presence of comorbidities on the clinical course of COVID-19 in infants was studied.

## 2. Material and Methods

The multicenter pediatric part of the national SARSTer database (SARSTer-PED) includes clinical and epidemiological data on children and adolescents (0 to 18 years) hospitalized in Poland with COVID-19 and diagnosed between 1 March 2020 and 31 December 2021. Fourteen Polish inpatient centers dedicated to pediatric patients with COVID-19 reported their consecutive cases using an electronic questionnaire. In this study, we extracted data on infants (0–12 months), which were available from 13 centers. Diagnosis of COVID-19 was based either on a positive real-time polymerase chain reaction (RT-PCR) or antigen testing on a nasopharyngeal swab performed in certified diagnostics laboratories. Epidemiologic data included known exposure to a person with confirmed SARS-CoV-2 infection (in the household or otherwise), the duration of symptoms before presentation, duration of hospitalization, and any chronic comorbidity. As immunization against COVID-19 for children younger than 5 years of age was introduced in 2022, the vaccination was not available for infants in the studied period and, thus, was not analyzed. We recorded all clinical symptoms present at the time of admission or during hospitalization. We divided the analyzed period into two parts: the first from 1 March 2020 to 31 December 2020 (which corresponded to the first and the second waves of pandemic, when original, wild SARS-CoV-2 dominated), and the second from 1 January 2021 to 31 December 2021 (corresponding to the third and fourth waves, with a dominance of B.1.1.7—alpha, and B.1.617.2—delta variants).

The clinical course of COVID-19 was defined as follows: grade 0—asymptomatic, when no complaints or symptoms were present, and no abnormalities were found on physical examination; grade 1—mild, when signs of upper respiratory tract infection were present, with or without fever and other complaints, but without pneumonia; grade 2—moderate, when pneumonia without hypoxemia was present; grade 3—severe, when pneumonia manifesting with dyspnea and oxygen desaturation <94% was present; and grade 4—critical, when acute respiratory distress syndrome, shock, or any organ failure occurred.

The indications for hospitalization were mainly based on the clinical status of the child; however, there was a trend for referring to a hospital for all infants below 6 months of age with confirmed SARS-CoV-2 infection. In addition, during the first months of the pandemic, testing for COVID-19 was available in Poland only in hospital settings; thus, children with suspected SARS-CoV-2 infection were referred to the hospital for confirmation of the infection, irrespective of their clinical presentation.

Statistical analysis was performed using MedCalc Statistical Software version 20.123 (MedCalc, Ostend, Belgium, https://www.medcalc.org, accessed on 10 November 2022). A two-sided *p*-value <0.05 was considered significant. Continuous variables were presented as numbers (percentages) or the medians with interquartile ranges (IQRs) and were compared using the Mann–Whitney test. Categorical variables were presented as numbers with percentages and were compared using the chi-square test.

Ethical Statement

This study was performed in accordance with the ethical standards presented in the 1964 Declaration of Helsinki and its later amendments. It was approved by the local ethics committee of the Regional Medical Chamber in Warsaw (No KB/1270/20; date of approval: 3 April 2020).

## 3. Results

### 3.1. Study Group

Among 2771 pediatric and 9458 adult patients with COVID-19 included in the SARSTer-PED database during 2020 and 2021, there were 940 infants (aged 0–12 months): 289 were hospitalized in 2020 and 651 in 2021 (Figure 1). Demographic and epidemiological characteristics of infants included in the study are presented in Table 1. There were no significant differences between the patients hospitalized in 2021 and 2020 concerning the median age, the proportion of hospitalized newborns (0–28 days), sex, source of infection, number of children with comorbidities, and duration of hospitalization (Table 1). However, fewer patients were hospitalized for over 7 days in 2021 compared to 2020 (16% vs. 22%, *p* = 0.009). The most commonly reported potential source of infection in infants was an infected family member (mainly a parent), which was reported in 54% of cases (52% in 2021 and 58% in 2020). Among the studied infants, 107 (11%) suffered from comorbidities, with cardiovascular diseases and prematurity as the most common (Table 2).

### 3.2. Clinical Presentation of COVID-19

The clinical symptoms of COVID-19 in the studied group are presented in Table 3. The most commonly reported symptom was fever (>38 °C), both in 2021 (64%) and 2020 (66%). Cough, rhinitis, loss of appetite, and dyspnea were more frequently reported among infants hospitalized in 2021 compared to those in 2020 (Table 3), whereas no significant differences were found concerning other reported symptoms. There were fewer asymptomatic children hospitalized in 2021 compared to 2020 (3% vs. 10%, *p* < 0.0001), which may be due to different indications for hospitalization in both these periods. In a total of 209 (22%) hospitalized children, pneumonia was diagnosed based on clinical evaluation or a chest X-ray examination, without significant differences among infants hospitalized in 2021 (21%) and 2020 (24%). One six-month-old infant hospitalized in December 2021 died due to respiratory failure. The child suffered from underlying diseases, including microsomia, hypotrophia, Hashimoto disease, and suspicion of interstitial lung disease. This one case makes the fatality rate 0.15% in 2021 and 0.11% overall.

### 3.3. Severity of COVID-19

The clinical course of COVID-19 in this cohort was most frequently described as mild (71%), both in 2021 (75%) and in 2020 (65%). However, there was a significant difference in the clinical course assessment between 2020 and 2021, with more severely ill infants in 2021 and more asymptomatic patients in 2020 (admitted mainly during the first weeks of the pandemic for epidemiological reasons; Table 4). In total, 4% of patients in our cohort were reported as severely or critically ill (grades 3 to 4), including 4% of the participants hospitalized in 2021 and 3% in 2020. We found a significant influence of the underlying comorbidities on the clinical course of the disease (Table 5). Infants with comorbidities were more prone to present with a severe or critical course of COVID-19 (grade 3 to 4) compared to patients without any comorbidity, which was observed for the whole cohort (10% vs. 3%, *p* = 0.0002) for infants hospitalized in 2021 (11% vs. 3%, *p* = 0.01) and in 2020 (9% vs. 2%, *p* = 0.01).

### 3.4. Treatment

Most infants received symptomatic treatment. Only three patients hospitalized in 2021 received the anti-SARS-CoV-2 treatment with remdesivir (Table 6). About one-third of patients (37% in 2021 and 30% in 2020; *p* = 0.03) received empirical antibiotics due to accompanying bacterial infections. Only a small proportion of infants required oxygen therapy (including high-flow nasal oxygen), mechanical ventilation, as well as treatment in the intensive care unit (Table 6).

## 4. Discussion

The COVID-19 pandemic causes a progressive increase in childhood morbidity worldwide. Many reports indicate that the clinical picture may vary depending on the circulating strain of the virus [10,11,12]. It has been proven that successive variants of SARS-CoV-2 differ in their infectivity and induce a specific clinical course of COVID-19 in all age groups. This study attempts to answer the question of whether infection with specific variants of SARS-CoV-2 determines its course in infants. We compared the epidemiology and clinical features of COVID-19 in infants hospitalized in 2020 and 2021 based on observations from 13 centers in Poland. Available epidemiological data show that the SARS-CoV-2 wild-type strain dominated in 2020, while VOCs emerged in 2021, with alpha (B.1.1.7) dominating in the first half of the year and delta (B.1.617.2) in the second half of 2021 [13]. In the retrospective analysis of 945 Ukrainian children hospitalized due to COVID-19 from April 2020 to February 2022, children aged 1–12 months predominated during wave IV caused by the delta variant [14].

Among the clinical manifestations of COVID-19 in infants, fever was predominant among both the children hospitalized in 2021 and those in 2020. Cough, rhinitis, and loss of appetite were significantly more frequently observed in 2021 (*p* < 0.0001). A similar course of COVID-19 in infants was described by Canadian authors [15]. The course of the disease was mild in most of the children we analyzed, although infants hospitalized in 2020 were more likely to have an asymptomatic course of SARS-CoV-2 infection, which could be due to the early period of the pandemic and slightly different indications for hospitalization.

In 3% of children hospitalized in 2021, the clinical course of the disease was classified as severe, in 1% as critical; in 2020, these numbers were 2% and 1%, respectively. These data are consistent with the observations of Canadian authors. Our study confirmed that severe and critical conditions were significantly more common in children with comorbidities, especially cardiovascular disease. Schober et al. also highlight the impact of comorbidities on the severity of COVID-19 in infants. Piche-Renoaud et al., based on an analysis of SARS-CoV-2 infection in 531 children, also highlight comorbidities and the younger age of children as risk factors for a more severe course of COVID-19 [15].

Chronic comorbidities pose a risk factor for the severe course of COVID-19 in both adult and pediatric patients and consequently increase hospitalization rates. Numerous meta-analyses showed a progressive increase in the number of hospitalizations of children with comorbidities during subsequent waves of the epidemic compared to the initial period of the pandemic [15,16]. Schober et al., in an age-adjusted multivariate logistic regression analysis, showed that comorbidities were one of the independent risk factors for a severe course of COVID-19 in children [5]. Age was not an independent risk factor, but various age-dependent comorbidities were associated with a more severe course of the disease in age-adjusted analyses: heart and lung disease in children <12 years of age and obesity in children >12 years of age. Among infants, neurological disorders and cardiac conditions were independent predictors of severe disease. In our study, the most common comorbidities included cardiovascular diseases and prematurity, then allergies, kidney diseases, lung diseases, genetically determined diseases, and gastrointestinal diseases. Their increase was also identified in the percentage of children hospitalized in 2021 compared to 2020. In 2021, immunosuppressive conditions, diabetes, and cancer were additionally observed among comorbidities. The high percentage of preterm births (21%) among children hospitalized in 2021 may reflect the impact of the pandemic on pregnant women and infections in this group caused by variants with higher infectivity. Preterm labor is one of the more common complications of SARS-CoV-2 infection in pregnant women, and the number of vaccinated pregnant women in 2021 was relatively low [17,18].

In our study, the number of hospitalized infants was significantly higher in 2021 than in 2020 (651 vs. 289, respectively). In contrast to the observations of Schober et al., there was no increase in the number of SARS-CoV-2 infections in newborns, which accounted for about 30% of hospitalized infants in the cited study [5]. It is estimated that about 7% of patients with COVID-19 in Poland require hospitalization [1]. However, data on the proportion of hospitalized children are unavailable. A study from the U.S. conducted between 1 March 2020, and 14 August 2021, identified 49.7 COVID-19-related hospitalizations per 100,000 children and adolescents. The highest rates were reported for children aged 0–4 years (69.2/100,000). During the period of delta variant dominance, the weekly rate of hospitalizations due to COVID-19 among children aged 0–4 years was almost 10 times higher compared to the period before delta dominance [19].

Our analysis showed similar lengths of infant hospitalization in 2020 and 2021, but significantly more children were hospitalized for longer than 7 days in 2020 (*p* < 0.009). Similar observations on the shortening of hospitalization were described by Ukrainian authors [14]. It seems that with the increase in the duration of the pandemic, some knowledge and experience in the management of patients infected with SARS-CoV-2 was acquired, which was reflected in the development of recommendations and, consequently, updating of routine clinical practice also with regard to indications and duration of hospitalization. In addition, at the onset of the 2020 pandemic, infants with COVID-19 with an asymptomatic course were also hospitalized for epidemiological reasons.

The dependence of infection prevalence in groups of the youngest children on specific SARS-CoV-2 variants was also confirmed by other authors regarding the Omicron variant wave [20,21]. Between December 2021 and February 2022, the overall seroprevalence in the U.S. increased from 33.5% to 57.7%, whereas in the group of children aged 0–11 years, it increased from 44.2% to 75.2%. As of February 2022, about 75% of children and adolescents had serologic evidence of prior SARS-CoV-2 infection, and about one-third had become seropositive as of December 2021. These findings illustrate the high rate of Omicron variant infection, especially among children [20]. Franczak et al., evaluating the seroprevalence of SARS-CoV-2 IgG antibodies in 686 children aged 2 weeks to 18 years hospitalized for reasons other than COVID-19, showed the presence of these antibodies in 392 (57%) children. As of December 2021, an increase in the percentage of children with positive anti-SARS-CoV-2 antibody titters was observed, to 87.5% of patients hospitalized in April 2022. 69% of the children with detected antibodies were under 5 years old. The study showed an increase in the percentage of those children during the fourth and fifth waves of COVID-19 in Poland caused by the delta and omicron variants, respectively. The vast majority of parents of the studied children had no knowledge of their offspring being infected with COVID-19, which may indicate an asymptomatic or mild course of the disease [21].

Our study group showed no significant differences in the gender distribution of hospitalized children, with a slight consistent predominance of boys (56% in both 2020 and 2021), which corresponds to Sierakova’s observations of Ukrainian children [14]. In both periods analyzed, the most common route of infection for infants was family contact, confirmed in a slightly higher percentage of infants hospitalized in 2020 (58% and 52%, respectively). These observations are also confirmed by Canadian authors, who, based on an analysis of 531 infants infected with SARS-CoV-2, confirmed family contact in 69% of infants hospitalized for COVID-19 [15].

In the treatment of the analyzed infants, mainly antipyretics were used, and patient hydration was carried out. Only three patients hospitalized in 2021 required remdesivir, 30 (5%) required oxygen therapy, 44 (7%) received systemic corticosteroids, and 141 (21%) received corticosteroid inhalation. One child required mechanical ventilation and transfer to the Intensive Care Unit. Azithromycin was used significantly less often than in 2020, which was due to the updated recommendations for the management of a child infected with SARS-CoV-2 [22].

This paper is one of the few studies of SARS-CoV-2 infections in infants available and, to our knowledge, the first one covering such an extensive material of 940 children. This research, however, has some limitations, such as the lack of determination of SARS-CoV-2 variants in samples from individual patients and the use of the data on the identification of individual strains from epidemiological surveillance. Another limitation is that the analyses were performed only in a group of hospitalized children.

Concluding, more infants were hospitalized during the period of dominance of VOCs, especially the delta variant, compared to the period of dominance of the wild strain, despite the fact that indications for hospitalization did not include hospitalization of asymptomatic patients in that period. The course of COVID-19 was mild in most cases, characterized mainly by fever and respiratory symptoms. The presence of comorbidities, particularly on the cardiovascular side and prematurity, was associated with a more severe course of the disease in infants.

## Figures and Tables

**Figure 1 jcm-12-02479-f001:**
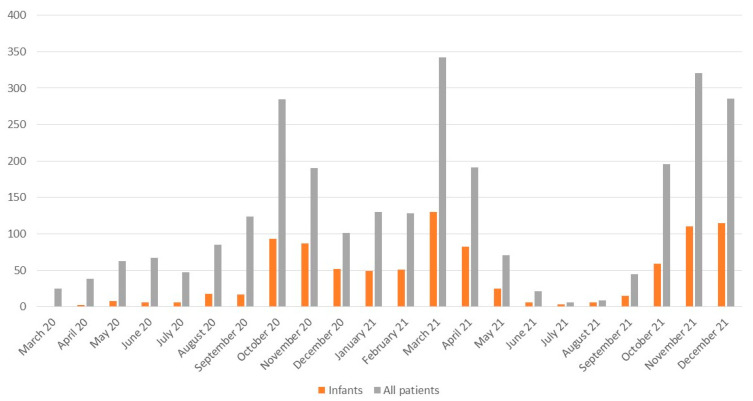
Number of infants diagnosed with COVID-19 in the subsequent months of 2020 and 2021 included in the analysis among the total number of hospitalized pediatric patients (0–18 years).

**Table 1 jcm-12-02479-t001:** Demographic and epidemiological characteristics of 940 infants included in the study.

Characteristics		Whole Study Group (*n* = 940)	Infants Hospitalized in 2020 (*n* = 289)	Infants Hospitalized in 2021 (*n* = 651)	*p*
Age (months)	Median (IQR)	4.0 (2; 7)	4.0 (1; 8)	4.0 (2; 7)	0.49
	Neonates	50 (5)	18 (6)	32 (5)	0.40
Sex	Male/Female	525 (56)/415 (44)	162 (56)/127 (44)	363 (56)/288 (44)	0.93
Contact with an infected person	Confirmed	522 (56)	170 (59)	352 (54)	0.17
	Houshold	505 (54)	168 (58)	337 (52)	0.06
	Other	17 (2)	2 (1)	15 (2)	
Comorbidity	Yes	107 (11)	34 (12)	73 (11)	0.80
Duration of symptoms before admission	Days, Median (IQR)	2 (1; 4)	2 (1; 3)	2 (1; 4)	0.0013
Duration of hospitalization	Days, Median (IQR)	4 (2; 7)	4 (2; 7)	4 (3; 6)	0.35
	Days, range	1–25	1–25	1–21	
	>7 days	166 (18)	65 (22)	101 (16)	0.009

Data are presented as numbers (%) or medians (IQR), respectively. IQR—interquartile range.

**Table 2 jcm-12-02479-t002:** Comorbidities among infants included in the study group.

Comorbidity	Whole Group (*n* = 940)	Children Hospitalized in 2020 (*n* = 289)	Children Hospitalized in 2021 (*n* = 651)	*p*
Any	107 (11)	34 (12)	73 (11)	0.80
Cardiovascular diseases	19 (2)	3 (9)	16 (22)	0.10
Prematurity	17 (2)	2 (6)	15 (21)	0.05
Allergy	16 (2)	3 (9)	13 (18)	0.22
Nephrological diseases	15 (2)	3 (9)	12 (16)	0.27
Respiratory tract diseases	13 (1)	4 (12)	9 (12)	0.93
Neurological diseases	10 (1)	6 (18)	4 (5)	0.04
Genetically determined diseases	9 (1)	3 (9)	6 (8)	0.91
Gastrointestinal diseases	6 (<1)	1 (3)	5 (7)	0.66
Immunosuppression	2 (<1)	0	2 (3)	1.0
Diabetes	2 (<1)	0	2 (2)	1.0
Oncological diseases	1 (<1)	0	1 (1)	1.0

Data are presented as numbers (%).

**Table 3 jcm-12-02479-t003:** Clinical symptoms of COVID-19 in the study group.

Symptom/Characteristics	Whole Group (*n* = 940)	Infants Hospitalized in 2020 (*n* = 289)	Infants Hospitalized in 2021 (*n* = 651)	*p*
fever	610 (65)	192 (66)	418 (64)	0.51
cough	482 (51)	110 (38)	372 (57)	<0.0001
rhinitis	424 (45)	96 (33)	328 (50)	<0.0001
loss of apetite	346 (37)	53 (18)	293 (45)	<0.0001
weakness	294 (31)	83 (29)	211 (32)	0.26
diarrhea	226 (24)	74 (26)	152 (23)	0.45
vomiting	104 (11)	28 (10)	76 (12)	0.37
dyspnea	85 (9)	16 (6)	69 (11)	0.01
rash	81 (9)	21 (7)	60 (9)	0.32
conjunctivitis	37 (4)	10 (3)	27 (4)	0.61
desaturation	35 (4)	7 (2)	28 (4)	0.16
seizures	14 (1)	5 (2)	9 (1)	0.68
asymptomatic	49 (5)	28 (10)	21 (3)	<0.0001
COVID-19-related pneumonia	209 (22)	70 (24)	139 (21)	0.33
death	1 (0.11)	0	1 (0.15)	0.50

Data are presented as numbers (%).

**Table 4 jcm-12-02479-t004:** The clinical course of COVID-19.

Grade	Whole Group (*n* = 940)	Infants Hospitalized in 2020 (*n* = 289)	Infants Hospitalized in 2021 (*n* = 651)	*p*
0	49 (5)	28 (10)	21 (3)	0.0004
1	674 (71)	189 (65)	485 (75)	
2	184 (20)	64 (22)	120 (18)	
3	28 (3)	7 (2)	21 (3)	
4	5 (1)	1 (1)	4 (1)	

Grade 0—asymptomatic; Grade 1—mild (without pneumonia); Grade 2—moderate (pneumonia without desaturation); Grade 3—severe (pneumonia with desaturation); Grade 4—critical (acute respiratory distress syndrome—ARDS, shock, organ failure). Data are presented as numbers (%).

**Table 5 jcm-12-02479-t005:** The clinical course of COVID-19 according to the presence of comorbidities.

Grade	Whole Group (*n* = 940)		Infants Hospitalized in 2020 (*n* = 289)		Infants Hospitalized in 2021 (*n* = 651)	
	With Comorbidity(107)	Without Comorbidity(833)	With Comorbidity(34)	Without Comorbidity(255)	With Comorbidity(73)	Without Comorbidity(578)
0	9 (8)	40 (5)	6 (18)	22 (9)	3 (4)	18 (3)
1	63 (60)	611 (73)	15 (44)	174 (68)	48 (66)	437 (76)
2	24 (22)	160 (19)	10 (29)	54 (21)	14 (19)	106 (18)
3	10 (9)	18 (2)	3 (9)	4 (1)	7 (10)	14 (2)
4	1 (1)	4 (1)	0	1 (1)	1 (1)	3 (1)
*p* value	0.0002		0.01		0.01	

Grade 0—asymptomatic; Grade 1—mild (without pneumonia); Grade 2—moderate (pneumonia without desaturation); Grade 3—severe (pneumonia with desaturation); Grade 4—critical (acute respiratory distress syndrome—ARDS, shock, organ failure). Data are presented as numbers (%).

**Table 6 jcm-12-02479-t006:** Treatment of COVID-19 implemented in the study group in 2020 and 2021.

Treatment	Whole Group (*n* = 940)	Infants Hospitalized in 2020 (*n* = 289)	Infants Hospitalized in 2021 (*n* = 651)	*p*
Empirical antibiotics	332 (35)	88 (30)	244 (37)	0.03
Azithromycin	56 (6)	48 (17)	8 (1)	<0.0001
Remdesivir	3 (<1)	0	3 (<1)	0.55
Convalescent plasma	1 (<1)	0	1 (<1)	0.50
Immunoglobulines	2 (<1)	0	2 (<1)	0.34
Steroids (systemic)	NA	NA	44 (7)	NA
Steroids (inhale)	NA	NA	141 (21)	NA
Oxygen therapy	36 (4)	6 (2)	30 (5)	0.06
High-flow nasal oxygen	2 (<1)	2 (<1)	0	0.03
Mechanical ventilation	1 (<1)	0	1 (<1)	0.50
Intensive care unit	6 (<1)	1 (<1)	5 (1)	0.45

Data are presented as numbers (%). NA—data unavailable.

## Data Availability

The datasets used and analyzed during the current study are available from the corresponding author upon reasonable request.

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
