# Peer review of "Clinical Course and Severity of COVID-19 in 940 Infants with and without Comorbidities Hospitalized in 2020 and 2021: The Results of the National Multicenter Database SARSTer-PED"

_jcm, 2023, doi:10.3390/jcm12072479_

Round 1
Reviewer 1 Report
The authors summarized hospitalized infants with COVID-19 in Poland, comparing those in 2020 to those in 2021.They found more hospitalized infants in 2021 by delta variant, and cardiovascular system and prematurity are comorbidities associated with severe diseases. Although the data are important, the figures and tables are abundant and discussions are out of focus.
Major comments
1. Lines 116-118; Could the authors estimate the proportion of the children and adolescents included in this study compared to the overall number of children and adolescents in the country?
2. In Material and Methods, the timing of introducing COID-19 vaccine including infants, children, and adults might be presented in order that the reader understand the national situation of COID-19 vaccine in Poland, instead of Lines 302-303 in Discussion.
3. The main topics include clinical manifestations, severity and comorbidities as described at the end of Introduction. However, the logic of the Discussion are disarranged and out of focus. The Discussion should be rewritten.
4. Generally, the main findings of the study are presented at the beginning of Discussion.
5. Lines 318-320; How many childhood populations are in Poland? Could the authors estimate similar numbers per 100,000 populations?
6. Lines 380-382; Does this support the finding of the prematurity being the risk for increased number of hospitalizations in 2021? If this hypothesis is true, the number and proportion of neonatal hospitalizations including both preterm and term neonates should be increased compared to the increased number of infants hospitalizations. However, it does not seem true in Table 1.
Minor comments
1. Lines 79-84; When the data derived from the same country of this study, it is better to describe the study site.
2. Line 99; It is no longer the “most” notable transmissibility for delta variant, because of the emergence of omicron variant.
3. Line 123; What is second-generation? Please add explanation or cite a reference.
4. Table 1 and 2; The number may be displayed with “n=” in the first row.
5. Figure 2 and Table 3; 13 items are duplicated and they may be removed. The residual 2 items may be added to the main text (actually, one death are already written in the text).
6. Figure 3 and Table 4; They show the same content. One of those may be removed.
7. Figure 4 and Table 5; Similarly, they show the same content. One of those may be removed.
8. Line 287; Were accompanying bacterial infections just suspected or established? Please clarify.
9. Lines 332-343; Clarifying the data in Poland would help readers understand the discussion.
10. Line 346; Please cite a reference.
11. Lines 358-359; I could not understand the logic of this sentence.
12. Lines 359-361; Why not move this sentence to the previous paragraph regarding indications for hospitalization?
13. Line 405; Please cite a reference.
Reviewer 2 Report
Manuscript is well written. The only weakness is the decreased actuality of COVID-19 at the moment. But there are stil many readers who are inetrested in COVID-19 topic, I think.
FRom my point of view, there is one point what should be explained. The most infants in the study had a mild course of disease, no severe symptoms. Some were even without smptoms. Why were these infants hospitalized? In Germany, for example, only people wit severe course were hospiralized , for example people who needed mechanical vantilation.
Round 2
Reviewer 1 Report
Thank you for respecting the Reviewer's comments.